# Homogenized Phylogeographic Structure across the Indo-Burma Ranges of a Large Monoecious Fig, *Ficus altissima* Blume

**Jian-Feng Huang** [1,2] , **Clive T. Darwell** [3] **and Yan-Qiong Peng** [1,2,*]

1   CAS Key Laboratory of Tropical Forest Ecology, Xishuangbanna Tropical Botanical Garden, Chinese Academy of Sciences, Mengla 666303, China; huangjianfeng@xtbg.ac.cn
2   Center of Plant Ecology, Core Botanical Gardens, Chinese Academy of Sciences, Mengla 666303, China
3   National Center for Genetic Engineering and Biotechnology (BIOTEC), 113 Thailand Science Park, Pahonyothin Road, Khlong Luang 12120, Thailand; ctdarwell@gmail.com
*   Correspondence: pengyq@xtbg.ac.cn

**Abstract:** As well as bountiful natural resources, the Indo-Burma biodiversity hotspot features high rates of habitat destruction and fragmentation due to increasing human activity; however, most of the Indo-Burma species are poorly studied. The exploration of plants closely associated with human activity will further assist us to understand our influence in the context of the ongoing extinction events in the Anthropocene. This study, based on widely and intensively sampled *F. altissima* across Indo-Burma and the adjacent south China ranges, using both the chloroplast psbA-trnH spacer and sixteen newly developed nuclear microsatellite markers (nSSRs), aims to explore its spatial genetic structure. The results indicated low chloroplast haplotype diversity and a moderate level of nuclear genetic diversity. Although limited seed flow was revealed by psbA-trnH, no discernible phylogeographic structure was shown due to the low resolution of cpDNA markers and dominance of an ancestral haplotype. From the nSSRs data set, phylogeographic structure was homogenized, most likely due to extensive pollen flow mediated by pollinating fig wasps. Additionally, human cultivation and human-mediated transplanting further confounded the analyses of population structure. No geographic barriers are evident across the large study range, with *F. altissima* constituting a single population, and extensive human cultivation is likely to have had beneficial consequences for protecting the genetic diversity of *F. altissima*.

**Keywords:** *Ficus*; Indo-Burma; genetic structure; human cultivation; nSSRs

## 1. Introduction

As one of the 34 identified global biodiversity hotspots [1], Indo-Burma, covering Burma, Thailand, Laos, Cambodia, and Vietnam, as well as parts of southern China, northeast India, and Bangladesh, harbours great diversity in both plants and animals. The endemic species' ecological and evolutionary trajectories were strongly influenced by plate tectonics, climatic oscillations, river system dynamics, sea level fluctuations, shifting coastlines, and human activity [2–5]. However, the biodiversity of the Indo-Burma region is severely threatened by factors such as human population growth, deforestation and habitat conversion, resource exploitation, pollution, and global warming [6–8]. Moreover, our knowledge about the underlying genomic structures (e.g., population structure, phylogeography, cryptic diversity) among Indo-Burma flora and fauna is extremely poor, mainly due to political instability and lack of infrastructure during long periods of the 20th century in many parts of this region [9,10].

In recent years, the conditions that prevented the researchers from entering large parts of Indo-Burma to explore the biodiversity have been improved along with the general development in this region. A few phylogeographic studies were conducted among

Indo-Burma fauna, including flukes [11,12], insects [13], fish [14,15], amphibians [16], reptiles [5], rodents [17,18], birds [9], and mammals [19]. For example, studies that focused on habitat-dependent freshwater animals revealed complex drainage re-alignments and sea-level fluctuation histories in the Indo-Burma region [5,15,20]. Drainage has also played an important role in shaping the phylogeography of plant species with limited pollen dispersal capability [10]. The population genetic investigation of the endemic species *Dalbergia cochinchinensis* Pierre and *D. oliveri* Gamble ex Prain represents the first detailed analysis of landscape genetics for tree species within Indo-Burma, although these two plants do not cover the entire Indo-Burma region [10]. Nonetheless, our knowledge about the genetic diversity and structure of Indo-Burma species is still extremely poor, especially for plants, which also impedes biodiversity conservation programs in this region.

The arrival of the Anthropocene has seen the transformation of ecosystems according to human use requirements [2,21]. Understanding how human activities affect the genetic diversity and structure of plants and animals is critical from both a scientific and policy perspective. Human-related threats, such as habitat destruction and fragmentation, invasive species, resource exploitation, and other indirect effects of human activities, often cause catastrophic biodiversity decline [22,23]. However, there are also some afforestation and ornamental plant species that are widely cultivated and play an integral role in the development of human civilization. Although their natural populations could have been reduced due to human-related threats, their genetic diversities may have been preserved due to extensive cultivation by humans. Few studies have focused on plants that are closely associated with humans, particularly those that inhabit badly degraded Indo-Burma forests. Studying the phylogeographic patterns of such species will be particularly beneficial to the ongoing biodiversity conservation of Indo-Burma flora and provide insight regarding the impact of human activity and whether it may degrade or augment genetic diversity.

With more than 800 species, *Ficus* L. (Moraceae) occurs pan-globally in both tropical and subtropical biomes. Coupled with their ecological significance as keystone species, and their typically species-specific relationship with co-evolved pollinating fig wasps (Hymenoptera, Chalcidoidea, Agaonidae) [24,25], the genus *Ficus* has long fascinated biologists. Many members of *Ficus* are also closely associated with humans, and often traditionally used as sources of medicines and food, as ornamental trees, religious plants, lac insect hosts (that exude useful lac gum products), fodder, fuel, hedges, or enclosures [26]. Among them, *F. altissima* is commonly known as the council tree and is frequently planted in cities, villages, or temples, both as ornamental and sacred plants [27]. The tree is also one of the recorded hosts of lac insect, and whose leaves and bark can be used as skin-disease treatments [28]. It is a long-lived perennial and hemiepiphytic tree (over 40 m tall), which can occupy a patch of about 300 m$^2$ (and thus almost constitute a single ecosystem [28,29]). *F. altissima*, as a keystone species, occurs naturally and widely in forests of tropical Asia at low densities and produces figs throughout the year in synchronous crops with asynchrony between trees [27,30].

*Ficus altissima* is a good example of a tree associated with many species and that is widespread in Indo-Burma and closely associated with human activity. Here, we present a phylogeographic study of *F. altissma* at a broad-scale across Burma, Thailand, Cambodia, Vietnam, and southern China. Based on an intensive sampling strategy across Indo-Burma, working with both chloroplast and nuclear DNA markers, we aimed to (1) investigate the level and distribution of genetic diversity of *F. altissima* within and among populations, as well as between populations collected from introduced and native areas; and (2) explore the population genetic structure of *F. altissima* within Indo-Burma. It will contribute to our understanding of the historical phylogeography of this keystone forest species of the Indo-Burma hotspot, as well as the influence on it from human activities.

## 2. Materials and Methods

### 2.1. Sample Collection

*Ficus altissima* populations were sampled in 37 locations in southern China, Burma, Thailand, Vietnam, and Cambodia (Figure 1, Table A1), yielding 267 individuals. The Fuz, Gan, Nid, Pan, Xia, and Yib populations were collected from areas outside its native range [31,32] and considered as introduced populations by human. However, within the native range, populations featuring disturbed habitats cannot be distinguished as either natural or cultivated. Populations such as Don and Chi, sampled from cities, may originate from their respective nearby wild source populations, rather than introduced from distant areas. The sample size for each population ranged from 1 to 23 individuals, according to limits of the population (Table A1).

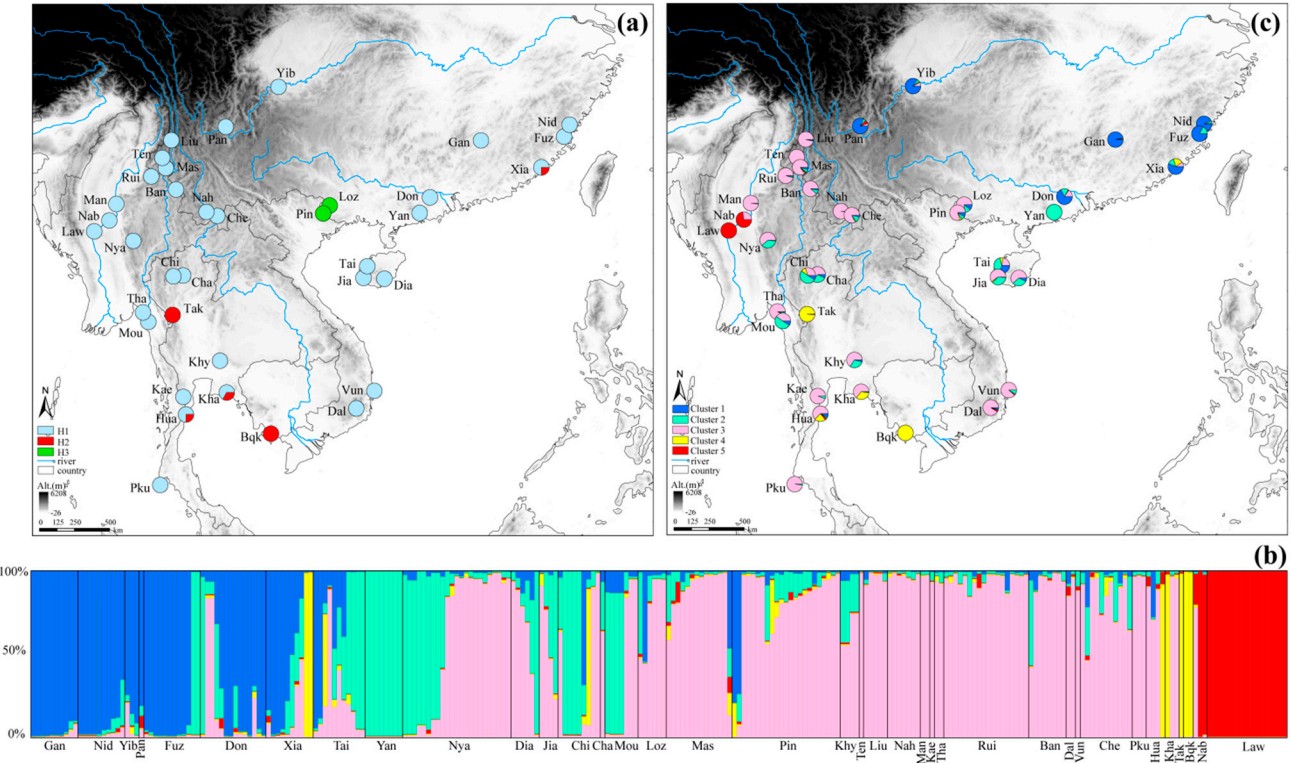

**Figure 1.** Sample locations and genetic structure of *F. altissima* in Indo-Burma and adjacent southern China: (**a**) geographical distribution of cpDNA haplotypes; (**b**) bar plots of the membership probabilities of *F. altissima* individuals to the different clusters from the STRUCTURE analysis at *K* = 5 based on 16 nSSRs; (**c**) geographical distribution of the genetic clusters detected in STRUCTURE, and the pie charts represent the assignment values of the admixed clustering analysis performed in STRUCTURE for all individuals.

### 2.2. DNA Extraction and Chloroplast DNA

Total genomic DNA of *F. altissima* was extracted from silica-gel dried material using the Plant Genomic DNAKit (Tiangen Biotech, Beijing, China). Ten chloroplast (cpDNA) intergenic regions, including ndhF-rpl32, psbA-trnH, psbB-psbF, psbC-trnS, psbJ-petA, trnD-trnT, trnF-trnV, trnL-trnF, trnQ-rps16, and trnS-trnG [33–35], were tested for amplification and sequence variation. Some of these markers are recognised as informative from earlier phylogenetic and phylogeographic research of *Ficus* plants [35–38]. To test the ten candidate cpDNA markers, one or two individuals were randomly selected from each sampled population for PCR amplification using universal primer pairs under the conditions described by Shaw et al. [33,34] and Vieira et al. [35].

Each fragment was bidirectionally sequenced by the Beijing Genomics Institute (Shenzhen, China). All forward and reverse strands were edited and assembled using the

program Sequencher 4.5 (GeneCodes, Ann Arbor, MI, USA). The sequences were aligned with Clustal X [39] and then adjusted manually using BioEdit 7.0.9.0 [40]. Ultimately, only psbA-trnH showed variation in preliminary tests and was selected for the PCR amplification of all individual samples.

### 2.3. nSSRs

In preliminary experiments, five nuclear DNA-based markers were tested, including ETS, ITS, GBSSI, G3pdh, and ncpGS [41,42]. However, all of them showed heterozygous sequencing peaks, suggesting clone contamination, and were subsequently abandoned. Alternatively, polymorphic nSSRs were developed. One sample of the population Nah was selected to construct the DNA library following the manufacturer's protocols of the KAPAHyper prep kit (KAPA Biosystems, Wilmington, MA, USA). The sequencing processes were performed by Microread Genetics Incorporation (Beijing, China) on HiSeq 2500 (Illumina, San Diego, CA, USA), using the corollary reagents from Illumina and with paired read lengths of $2 \times 150$ base pairs and yielding 3.71 G raw bases. Raw data were then quality filtered by trimming the adapter sequences and by removing reads with quality scores < 20. High-quality reads were assembled into 15,532 contigs. The software MIcroSAtellite [43] was used to identify contigs with nSSRs, and Primer 3.0 [44] was used to design microsatellite primers using the default settings. Eighty primer pairs were selected as candidates for amplification and polymorphism testing. Finally, 16 of the 80 tested primer pairs were found to produce repeatable amplicons with high polymorphism and were used for genotyping. The Multiplex PCR amplification was conducted in 10 µL volumes mixed with 2.0 µL of $5 \times$ Buffer, 0.3 µL of dNTP (10 mM each), 1 µL of DNA (100 ng), 0.4 µL of primer mixture (5 µM), and 0.1 µL of Taq polymerase (5 U), based on the following the conditions: initial denaturation at 95 °C for 8 min, 30 cycles at 94 °C for 30 s, at 56 °C for 30 s, at 72 °C for 30 s, and at 60 °C for 30 min for the final extension. Post-PCR products were analysed by capillary electrophoresis on an ABI 3730XL DNA analyser (Applied Biosystems, Foster City, CA, USA) with the GeneScan 500 ROX Size Standard. Microsatellite fragment sizes were determined using GeneMapper version 3.2.

### 2.4. Confirming Usability of nSSRs

Micro-checker v.2.2.3 [45] was used to examine the presence of any genotypic errors due to stuttering, large allele dropout, and null alleles. Linkage disequilibrium (LD) and Hardy–Weinberg equilibrium (HWE) were tested by GENEPOP 4.7.0 [46]. Significance levels were corrected using the Bonferroni correction for multiple tests.

### 2.5. Genetic Diversity

The cpDNA haplotypes were distinguished using DnaSP v5 [47] on the basis of nucleotide and indel differences. Molecular diversity indices, including the number of haplotypes ($H$) as well as nucleotide ($\pi$) and haplotype ($H_D$) diversity, were calculated using DnaSP v5. For the nSSRs data, classical indices of genetic diversity were estimated using GenAIEx 6.5 [48]. The inbreeding coefficient ($F_{IS}$) was calculated by FSTAT 2.9.3 [49]. The polymorphism information content (PIC) for each nSSRs was calculated with Cervus v3.0.7 [50,51]. Subsequently, the mean number of alleles ($N_a$) and private alleles ($PA_r$) per locus, observed ($H_O$), and expected heterozygosity ($H_E$) of the 37 sampled populations based on nSSRs were used to show the geographic pattern by using the inverse distance weighted (IDW) interpolation function implemented in ArcGIS 10.3 (ESRI, Redlands, CA, USA). IDW assumed that points close to each other are more relevant than those that are more distant and is weighted more closely to the predicted position than the farther distances [52].

### 2.6. Genetic Structure

Genetic differentiation based on cpDNA and nSSRs data were evaluated by analysis of molecular variance (AMOVA) performed in Arlequin v3.5 [53], with 10,000 permu-

tations used for tests of significance. For nSSRs data, a pattern of isolation by distance (IBD) was assessed using a Mantel test in GenAlEx 6.5 with 9999 permutations to evaluate the correlations between pair-wise genetic ($F_{ST}/(1 - F_{ST})$) and geographic distance. The geographic distances between pairwise populations were calculated using the program Geographic Distance Matrix Generator v1.2.3 [54] based on their decimal degree coordinates. To estimate the interpopulation genetic affinity of studied populations, principal component analysis (PCoA) was conducted with GenAlEx 6.5 based on Euclidean distance and an UPGMA phylogenetic tree was constructed by MEGA 6.06 [55] based on Nei's genetic distance. To further understand the clustering patterns, the Bayesian clustering of individual genotypes was investigated using STRUCTURE v2.3.4 [56]. We employed a model with admixture, with a burn-in period of 100,000 and a run length of 1,000,000 iterations varying *K* from *K* = 1 to *K* = 10. For each value of *K*, ten runs were done. The STRUCTURE HARVESTER online program [57] was used to detect the optimal *K* value using Evanno method [58]. CLUMPP 1.1.2 [59] was used to summarize the membership coefficients into clusters.

### 2.7. Population Dynamics

For cpDNA, the population expansion hypothesis was tested using neutrality and mismatch distribution tests. For the neutrality test, Tajima's *D*, considering the frequency of mutations [60], and Fu's $F_S$ [61], based on cpDNA haplotype distribution, were conducted. The mismatch distribution test is used to assess whether the observed distribution of the pairwise differences matches the expectations under the sudden demographic expansion and the spatial-demographic expansion models. These analyses were conducted in Arlequin v3.5.

The recent demographic bottlenecks were tested for the populations with more than ten individual samples using the program BOTTLENECK 1.2.02 [62] with two recommended evolutionary models for nSSRs data [63,64]: the stepwise mutation model (SMM) and the two-phase mutation model (TPM). Under the TPM, the proportion of SMM and IAM (infinite alleles model) were set with default values that 70% of the mutations were assumed to occur under the SMM, and 30% of the mutations were assumed to occur under IAM. For each mutational model, 10,000 replicates were performed. After a reduction in the effective population size, the allele diversity is expected to drop more rapidly than their heterozygosities drop [65]. Thus, the heterozygosity of the population is larger than expected considering the number of alleles found. Sign test and the allele frequency distribution mode shift analysis [66] in BOTTLENECK 1.2.02 were performed to determine the population genetic reduction signatures characteristic of recent reductions in effective population size.

## 3. Results

### 3.1. cpDNA Genetic Diversity

Of the ten tested cpDNA intergenic regions, only psbA-trnH (356 bp) showed sequence variation in *F. altissima*. Of the 233 obtained sequences, only three haplotypes (GenBank accession numbers: MT291815–MT291817) were detected with two indels and two single nucleotide polymorphisms, suggesting limited use at the population level. At the population level, $H_D$ and $\pi$ ranged from 0.000 to 0.667 and (0.000 to 0.495) $\times 10^{-2}$, respectively. Only three populations, Hua, Kha, and Xia, showed a haplotype polymorphism possessing two haplotypes, H1 and H2 (Table A1, Figure 1a). Haplotype H3 was derived from the ancestral common haplotype H1 by a single 13-bp indel. Haplotype H2 is different from H1 by two transversions and a 2-bp indel.

### 3.2. Usability of nSSRs and Genetic Diversity

Sixteen polymorphic nSSRs were successfully developed based on high-throughput sequencing and microsatellite genotype data for all 267 individual samples were obtained. They displayed relatively high polymorphism across *F. altissima* individuals. The sequences

are deposited in GenBank and detailed characteristic information of these 16 nSSRs is listed in Table 1. The PIC values ranged from 0.137 for the Fal_61 locus to 0.813 for the Fal_73 with an average of 0.539. There was no evidence of stuttering errors or large allele dropout in any of the loci analysed by Micro-checker v.2.2.3. In turn, null alleles were detected at five loci in a few populations (Table A2). Across the 16 loci and 37 studied populations, 136 microsatellite alleles were identified, corresponding to 8.5 alleles per locus and ranging from 5 to 12 alleles for individual loci. A few pairs of loci among a few populations exhibited linkage disequilibrium. Some populations deviated from HWE in a few loci, and a deficiency of heterozygotes was observed by GENEPOP in these populations, which may be due to the presence of null alleles (Table A2). The genetic diversity parameters for *F. altissima* populations are summarized in Table A1, with an observed species-level heterozygosity ($H_O$) of 0.483, expected heterozygosity ($H_E$) of 0.576, and a species inbreeding coefficient ($F_{IS}$) of 0.163. For each population, the $H_O$, $H_E$, and $F_{IS}$ ranged from 0.063 to 0.563, 0.031 to 0.580, and $-1.000$ to 0.451, respectively. In turn, the number of private alleles ($N_p$), mean number of alleles ($N_a$), and private alleles per locus ($PA_r$) ranged from 1 to 3, 1 to 5.125, and 0.000 to 0.188, respectively. Populations Hua, Pin, and Xia each have three private alleles and showed the highest $PA_r$ values. However, no private alleles were found in 22 of the 37 sampling localities. Most of the loci showed negative or small positive $F_{IS}$ values, which is consistent with the prevalence of outcrossing in monoecious fig species. The values of $N_a$, $PA_r$, $H_O$, and $H_E$ are geographically displayed in Figure 2. The regions with high levels of genetic diversity were scattered across the Indo-Burma hotspots and adjacent southern China. Unexpectedly, the six introduced populations showed similar levels of genetic diversity with the populations collected from the native range, and many populations collected from disturbed urban areas, such as Mas, Rui, Don, Pin, and Chi, also showed high levels of genetic diversity (Figures 1 and 2, Table A1).

**Table 1.** Sixteen nSSRs developed for *F. altissima*.

| Locus | Primer Sequences | Repeat Motif | GenBank Accession No. | PIC |
|---|---|---|---|---|
| Fal_2 | F: CCTGTGGGAGAGTTTGAAGG R: CTTGCTGCACGAATCTGCT | (CGT)6 | MN255360 | 0.487 |
| Fal_9 | F: GAGTACATGCAAATGCCTCG R: CTCAGCAGCAACGAAAGATG | (TTTA)5 | MN255361 | 0.561 |
| Fal_11 | F: GACCTGTTGGAGGAGATTGC R: TCATGGGCCACTTATCCTTC | (GCG)5 | MN255362 | 0.753 |
| Fal_14 | F: CGATCCTTATCCTCTGCTCG R: GCACGCAATTTGAACGAAC | (AAG)10 | MN255363 | 0.740 |
| Fal_28 | F: TCAGAATTGGAACGAGGGAC R: GCAGGGACTTCTTCTCTGACC | (TTTG)7 | MN255364 | 0.492 |
| Fal_31 | F: CGATCACCACGAGCTACTGA R: TGGCGCATGATAAGTTTGAG | (GTCT)5 | MN255365 | 0.803 |
| Fal_34 | F: CCAACTAGCCACACTTTGGA R: TGGCACAATTGACCTCAGAA | (AATCCC)5 | MN255366 | 0.582 |
| Fal_41 | F: CTCTTGGATACCGAGTCCGA R: GGACTGAACTGCTGTCATGTG | (TTTG)5 | MN255367 | 0.434 |
| Fal_45 | F: TCGAAATCGGATACTCCTCG R: CATGAAGCTTGAGCATTGGA | (TTTTAT)5 | MN255368 | 0.583 |
| Fal_46 | F: GCCACGACATCACATCATTA R: TCAGCTTACCTTATTGGCCG | (ACAT)6 | MN255369 | 0.444 |
| Fal_48 | F: ATGTGCCAAACCCAGAACTC R: CAACCTAGCTCTCGGAGGTG | (AAAC)5 | MN255370 | 0.764 |
| Fal_50 | F: GCCCATCTGGTGACTGAAAC R: CGTGTGCATGCTTCATCTCT | (AAT)7 | MN255371 | 0.351 |
| Fal_61 | F: TGGGCTCGTGACTGACTAGA R: ATGTGGGGACGGCCTCTT | (TTG)6 | MN255372 | 0.137 |
| Fal_62 | F: CACGTGGTGGCTATGTTCTG R: GCTACGGTTTATTTGCGGTG | (TTA)7 | MN255373 | 0.215 |

**Table 1.** *Cont.*

| Locus | Primer Sequences | Repeat Motif | GenBank Accession No. | PIC |
|---|---|---|---|---|
| Fal_73 | F: ATCCTTTGCTTTGCTCGTGT | (ATA)9 | MN255374 | 0.813 |
|  | R: CGAACCTTGCACACCCTAAT |  |  |  |
| Fal_75 | F: GGATCCAAAATTGGGCAGT | (AAT)8 | MN255375 | 0.459 |
|  | R: ATTCATGGAATCATGGGCAC |  |  |  |

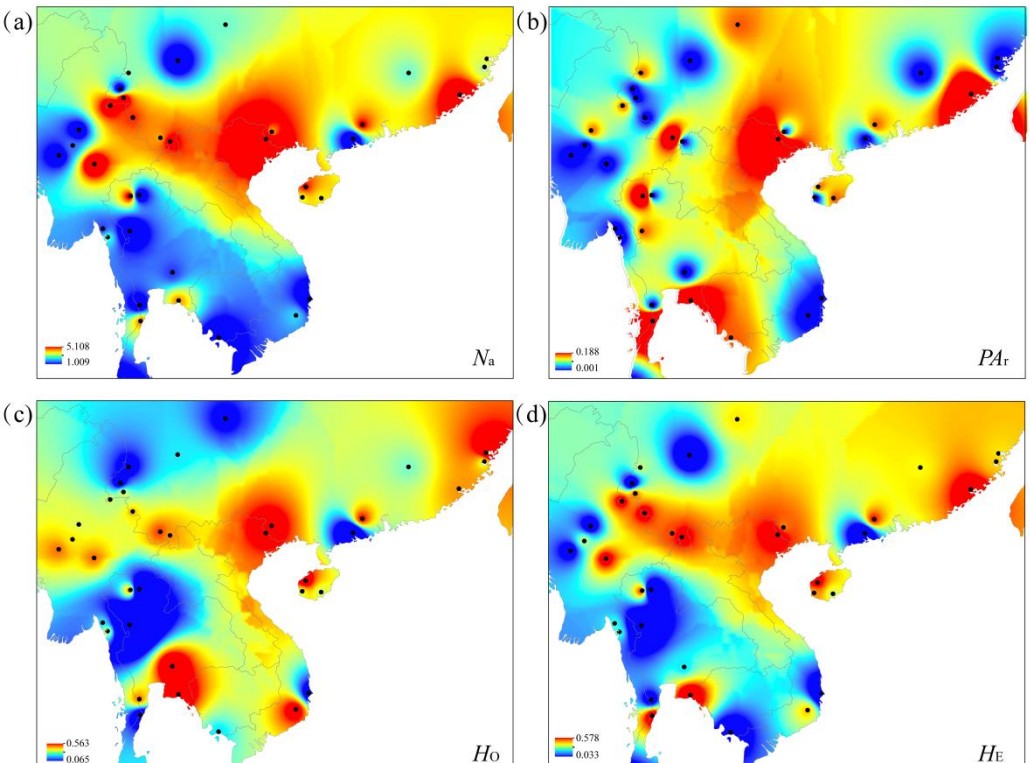

**Figure 2.** Genetic diversity maps of the 37 *F. altissima* populations (black dots) in Indo-Burma and adjacent southern China: (**a**) IDW interpolation of the mean number of alleles per locus ($N_a$); (**b**) IDW interpolation of the mean number of private alleles per locus ($PA_r$); (**c**) IDW interpolation of the observed heterozygosity ($H_O$); (**d**) IDW interpolation of the expected heterozygosity ($H_E$).

*3.3. Population Structure*

Chloroplast haplotype H1 was extremely dominant in the whole sampling area, while population Xia shared haplotype H2 with distant Thailand populations Tak, Kha, and Hua. Haplotype H3 was endemic to Guangxi of China (Loz and Pin) (Figure 1a). AMOVA indicated strong differentiation among populations in cpDNA sequences for *F. altissima* ($F_{ST} = 0.958$, $p < 0.001$) and showed that genetic variation mainly occurred between populations (Table 2).

Based on the 16 nSSRs, pairwise $F_{ST}$ among populations ranged from −0.143 between populations Dal and Kae (979 km apart) to 0.804 between Yan and Tak (1468 km apart). Slightly negative pairwise $F_{ST}$ (i.e., treated as zero) occurs in several population pairs, suggesting that population differentiation is negligible. Such negative estimates were converted to zero in the IBD analysis. There is no obvious correlation between genetic and geographic distance ($p = 0.277$, Figure 3). AMOVA tests revealed that global $F_{ST}$ was low at 0.178 ($p < 0.001$), and most genetic variation (82.25%) is contained within populations, with only 17.75% genetic variation observed among populations (Table 2).

**Table 2.** Analysis of molecular variance (AMOVA) of the genetic diversity in *F. altissima*.

| | Source of Variation | df | SS | VC | PV (%) | $F_{ST}$ |
|---|---|---|---|---|---|---|
| | Among populations | 36 | 305.621 | 1.37 | 95.83 | |
| cpDNA | Within populations | 196 | 11.667 | 0.06 | 4.17 | |
| | Total | 232 | 317.288 | 1.43 | | $F_{ST}$ = 0.958 |
| | Among populations | 36 | 554.386 | 0.82 | 17.75 | |
| nSSRs | Within populations | 497 | 1888.006 | 3.80 | 82.25% | |
| | Total | 533 | 2442.391 | 4.62 | | $F_{ST}$ = 0.178 |

Notes: df, degree of freedom; SS, sum of squares; VC, variance component; PV, percentage of variation.

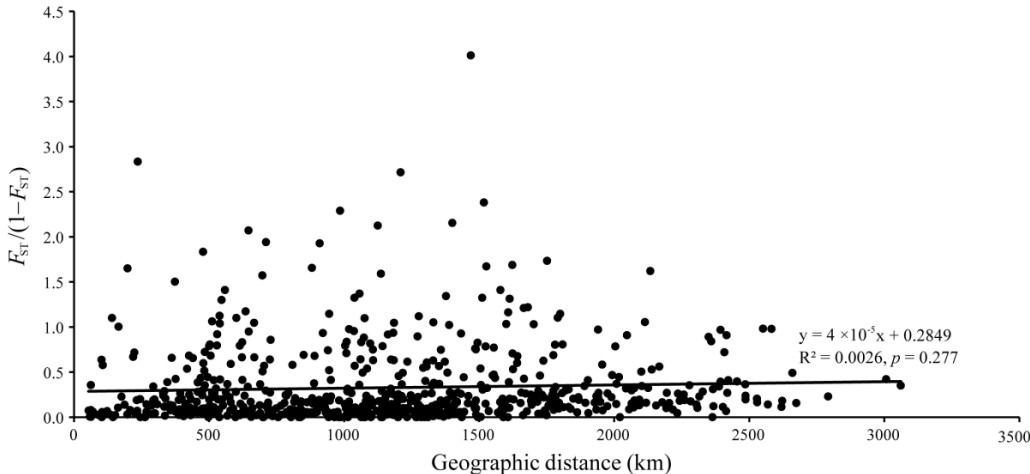

**Figure 3.** The regression of paired $F_{ST}/(1 - F_{ST})$ vs. the geographic distances was not significant for nSSRs data.

The STRUCTURE results of the nSSRs data indicated an optimal *K* value of 5 using the delta*K* criterion [58], but this does not correspond with any obvious biological/geographical interpretation. The bar plots of the membership probabilities of *F. altissima* individuals to the different clusters using STRUCTURE analysis for *K* = 5 is shown in Figure 1b. The geographical distribution of the genetic clusters is shown in Figure 1c and pie charts represent the assignment values of the admixed clustering analysis performed in STRUCTURE for all individuals. There are no obvious boundaries defining the adjacent clusters or populations. Even geographically distant populations are highly genetically homogeneous (Figure 1b,c). The lack of a clear phylogeographic structure for *F. altissima* across the Indo-Burma range, revealed by Bayesian population structure analysis, was further validated by the PCoA (Figure 4a) and UPGMA phylogenetic analyses (Figure 4b). The different populations collected in a country also do not cluster together (Figure 4).

*3.4. Population Dynamics*

Neither Tajima's *D* (−0.864, *p* > 0.1) nor Fu's $F_S$ (10.589, *p* > 0.1) vary significantly from zero, suggesting there is no evidence of recent population expansion or no recent bottlenecks [60,61]. The mismatch distribution analysis failed, as the least-square procedure to fit expected and observed mismatch distribution did not converge after 2000 steps. Based on the nSSRs data, under the TPM assumptions it was significant for five populations (Don, Fuz, Gan, Law, and Mas) while three populations (Gan, Law, and Nya) were significant under SMM (*p* < 0.05) and four populations (Gan, Law, Nid, and Xia) revealed a shifted mode of allele frequency distribution. However, only population, Law, was detected to have experienced a bottleneck effect by all of the three approaches (Table 3), and this population was genetically homogenous, as revealed by STRUCTURE analyses (Figure 1b).

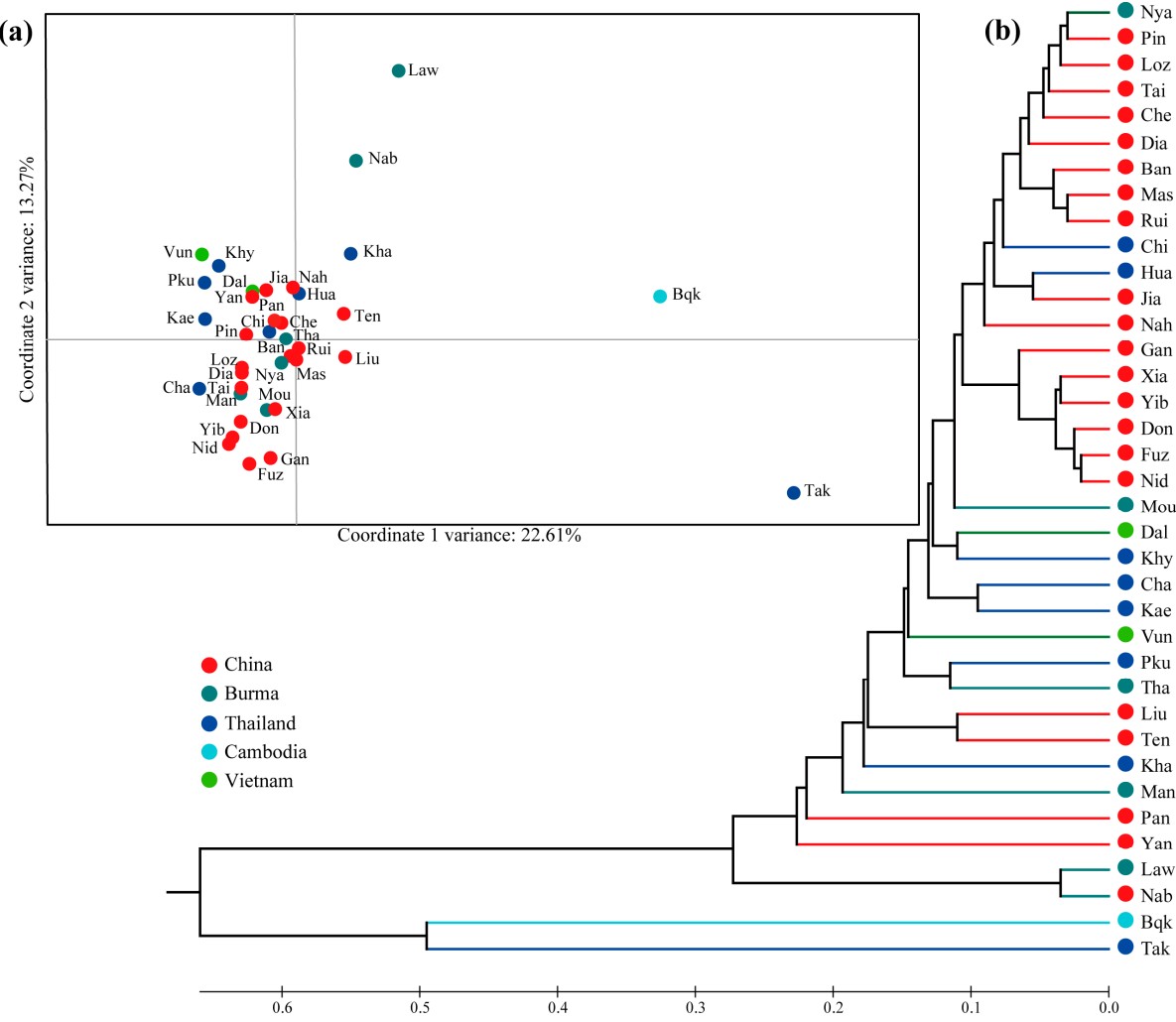

**Figure 4.** The interpopulation genetic affinity of 37 *F. altssima* populations based on microsatellite data: (**a**) two-dimensional scatter diagram based on principal coordinate analysis; (**b**) dendrogram based on the unweighted pair-group method with arithmetic average (UPGMA). The populations collected from different countries are marked in different colours.

**Table 3.** Bottleneck test of *F. altissima* populations with more than ten individual samples.

| Population | Size | Model Shift | TPM | SMM |
|---|---|---|---|---|
| | | | *p* Value | *p* Value |
| Che | 11 | normal L-shaped distribution | 0.337 | 0.448 |
| Don | 14 | normal L-shaped distribution | 0.047 * | 0.336 |
| Fuz | 12 | normal L-shaped distribution | 0.033 * | 0.245 |
| Gan | 10 | shifted mode | 0.010 * | 0.073 |
| Law | 17 | shifted mode | 0.002 ** | 0.005 ** |
| Mas | 14 | normal L-shaped distribution | 0.031 * | 0.0002 |
| Nid | 10 | shifted mode | 0.233 | 0.503 |
| Nya | 23 | normal L-shaped distribution | 0.366 | 0.008 ** |
| Pin | 23 | normal L-shaped distribution | 0.540 | 0.024 * |
| Rui | 18 | normal L-shaped distribution | 0.355 | 0.078 |
| Tai | 11 | normal L-shaped distribution | 0.133 | 0.566 |
| Xia | 10 | shifted mode | 0.551 | 0.075 |

Notes: * Significant deviation from mutation-drift equilibrium at $p < 0.05$; ** Significant deviation from mutation-drift equilibrium at $p < 0.01$.

## 4. Discussion

### 4.1. Genetic Diversity, Population Dynamics, and Phylogeographic Patterns of Ficus altissima

The present study revealed a much lower level of cpDNA haplotype diversity in *F. altissima* than for the other published monoecious *Ficus* species [36,38,67]. To screen polymorphic markers, ten commonly used cpDNA intergenic regions in plants with the most potentially informative characters [33,34,68] were employed in our preliminary experiments, but only psbA-trnH showed sequence variation. The psbA-trnH has been commonly used in fig species [36,38,69]. However, only three psbA-trnH haplotypes were distinguished from 233 individual samples. Thus, no discernible cpDNA phylogeographic structure was revealed due to the low resolution of cpDNA regions and dominance of haplotype H1 across the whole sampling range.

Haplotypes H1 and H2 were also found in closely related fig species. For example, H1 was shared by *F. benghalensis* L., and H2 was shared by *F. binnendijkii* Miq., *F. curtipes* Corner, *F. drupacea* Thunb., *F. microcarpa* L., and *F. retusa* L., suggesting the retention of ancient polymorphisms in these closely related species. The sharing of ancestral haplotypes among widespread monoecious fig species may be a fairly common phenomenon, as observed in F. altissima and F. insipida subsp. indipida Willd [36]. These species often have large effective population sizes and long generation times, which will prolong the evolution of reciprocal monophyly [36,70]. Despite low cpDNA diversity across all samples, AMOVA revealed high interpopulation differentiation for cpDNA, suggesting that seed dispersal of *F. altissima* is limited. The derived haplotype H3, only found in Guangxi, indicates low seed dispersal as it has not dispersed into surrounding regions, or is a very recent mutation. Although there is no detailed published information, birds are probably the seed dispersal agents of *F. altissima* as it possesses a typical bird-dispersal syndrome [71]. Indeed, small non-migratory birds, including *Megalaima* Gray (Capitonidae), *Dicaeum* Cuvier (Dicaeidae), and *Pycnonotus* Boie (Pycnonotidae) species, were observed feeding on *F. altissima* trees during our field work. *F. altissima* is a hemiepiphytic tree and it exists initially as an epiphyte. The seeds of *F. altissima* germinate in the canopy of host trees [72], rather than in the soil. Hence, the germination and early seedling survival appear to be dependent on microsites provision by the host trees, such as host diameter at breast height, height and position of colonization in the canopy of host trees, crown illumination, moisture retention, and slope angle [72]. The requirements for host taxa and microsites may hinder the dispersal of *F. altissima* seeds to colonize new habitats. Short-distance seed dispersal coupled with dependency on microsites for seed development could result in limited seed flow. Nonetheless, introduced population Xia collected from Xiamen city of China shared an ancestral chloroplast haplotype H2 with the distant Thailand populations (Tak, Kha, and Hua), suggesting that human-mediated long-distance dispersal may also play a role on the current species phylogeography.

Sixteen polymorphic nSSRs for F. altissima were developed based on high-throughput sequencing and revealed a moderate level of genetic diversity. Twenty-two of the 37 populations have no private alleles, suggesting strong interpopulation pollen flow, further supported by AMOVA analyses. Extensive long-distance dispersal of pollinating wasps, aided by passive air column travel [73,74], has been documented repeatedly in monoecious figs and is ranked among the furthest known in plants [75–77]. For example, Ahmed et al. [77] showed successful pollination across distances over 160 km in monoecious *F. sycomorus* L. Meanwhile, synchronous intra-tree flowering, between tree asynchrony [78], and low densities of monoecious figs mean host-specific pollinators must travel long distances. Most populations showed negative or small positive $F_{IS}$ values (Table A1), which is consistent with regular outcrossing associated with high dispersal events.

Rapid range expansion or a bottleneck event is not supported by assessment of Tajima's $D$ and Fu's $F_S$ from the cpDNA markers. However, this contrasts with the nSSRs data, where three populations (Don, Law, and Mas) were detected to have undergone bottleneck events (Table 3). However, Don is certainly a cultivated population. Mas and

Law were also collected from locations near human settlements with traces of cultivation, suggesting they experienced artificial bottlenecks.

Range expansions often result in reduced genetic diversity compared with the source populations, for example, glacial refugia compared to edge populations. Across our study range, no such significant changes in genetic diversity (e.g., haplotype diversity, observed heterozygosity, private allelic richness; Figure 2 and Table A1) based on nSSRs or cpDNA data were detected in *F. altissima*. Range expansion in obligate mutualisms involving free-living organisms is complicated by requiring the successful range extension of a pair of independently dispersing species [79,80]. Thus, the successful colonization and reproduction of *F. altissima* in a new location also depends on successful population establishment of the host-specific pollinating fig wasps, and pollinator absence often appears to limit range expansion of the *Ficus* host [80,81]. Furthermore, as a hemiepiphytic fig, germination and early seedling survival of *F. altissima* appear to be dependent on microsites provision by host trees [72]. This would be an additional impediment for the range expansion of *F. altissima*.

The origin of F. altissima has been estimated to be no earlier than the Upper Miocene [25,82], when Southeast Asia remained as a single rainforest block [83,84]. This is further supported by our finding of the widespread persistence of the ancestral H1 haplotype, across a contemporary range that is hypothesized to have undergone repeated replacement by savanna habitat during the Quaternary glaciations, with rainforest areas shrinking considerably and mostly persisting in lowland glacial refugia [85,86]. We assume that *F. altissima* survived in many lowland forests during Quaternary glaciations, but that postglacial expansions were slow. The subsequent short evolutionary history coupled with long generation times would have produced low cpDNA haplotype diversity, while genetic drift may have caused the loss of rare cpDNA haplotypes.

Indo-Burma has a complex geological and climatic history, as well as diverse habitats largely derived from the wide variation in landform, climate, and latitude [9,13,16]. Moreover, tropical forest tree species typically occur in low densities [87], and thus they may be more susceptible to genetic drift [88]. Therefore, there are both historical and ecological reasons to expect high levels of population differentiation and clear phylogeographic structure in forest trees of Indo-Burma [89,90], as observed in *D. cochinchinensis* and *D. oliveri* [10]. However, this study revealed a pattern of homogenized phylogeographic structure for *F. altissima* across Indo-Burma. As a relative generalist species, *F. altissima* is continuously distributed throughout Indo-Burma and occurs up to elevation of 2000 m. There are no obvious geographical barriers to impede the long-distance dispersal of *F. altissima*'s pollinators. In general, such postulated extensive pollen flow can buffer the effects of fragmentation and homogenize discrete populations [91,92]. A lack of phylogeographic structure for *F. altissima* in Indo-Burma appears to mainly derive from long-distance pollen dispersal. It differs from dioecious figs in which isolation by distance is often present within regions [93–95], but is consistent with patterns documented in large monoecious *Ficus* species with continuous habitat, as observed in monoecious *F. racemosa* L. [75] and *F. sur* Forssk. [96]. The notably reduced population differentiation in nuclear DNA compared to plastid cpDNA identified in *F. altissima* is also a common pattern documented in both monoecious [38,75] and dioecious fig species [94,96]. This pattern of homogenized phylogeographic structure for *F. altissima* may be observed among other Indo-Burma forest species with extensive pollen flow and limited seed transfer. However, large-scale anthropic cultivation confounds the analyses of the genetic structure of *F. altissima*.

### 4.2. Human Influence

Human influence on planet Earth is well-documented and considered as influential as natural processes in sculpting contemporary diversity patterns [97]. The rapid and massive loss of biodiversity directly caused (e.g., logging, hunting, fishing) and indirectly induced (e.g., pollution, infrastructure, greenhouse gas emission) by human activity in the Anthropocene is leading to the sixth mass extinction event [2,98–100]. Cultivated

populations often show evidences of a reduction in genetic diversity due to artificial selection for specific quality traits [101–104]. *F. altissima* is frequently planted in its native areas or introduced into non-native areas as sacred or ornamental trees without artificial selection of specific traits; however, wild populations are being heavily impacted. It is likely to have contributed to avoiding drastic decreases in *F. altissima* genetic diversity due to declines in wild populations. For example, almost every Dai ethnic village in Xishuangbanna of China transplanted *F. altissima* from neighbouring wild resources for cultural and religious purposes [105]. Our results showed relatively high genetic diversity in many populations sampled from Dai villages (e.g., Nah, Che, Chi) and disturbed urban areas (e.g., Mas, Rui, Don, Pin). The six introduced populations also showed similar levels of genetic diversity with the populations collected from native areas, suggesting that they may originate from more than one source. This study suggests that conservation by cultivation is an effective means for protecting the genetic diversity of *F. altissima*.

It is difficult for hemiepiphytic *F. altissima* seeds to germinate and grow in urban habitats far from the natural range due to the lack of suitable host trees. Given Xiamen city is a regional center with well-established ex-patriate links to Southern Asian countries and *F. altissima* is of great ornamental value, the unexpectedly discontinuous distribution of haplotype H2 between population Xia sampled from Xiamen city and populations Tak, Kha, and Hua sampled from Thailand could be explained by long-distance human transportation, rather than by natural seed dispersal agents.

The influence of human activity also most likely explains the random and chaotic clustering revealed by nSSRs data. For example, the population Law features uniform tree age among a highly disturbed habitat close to human habitation, while population Yan is a reforestation area. The uniform genetic constitution among different individuals within these populations (Figure 1b) may originate from an individual derived germplasm. Genetic bottleneck signals were detected for all the four cultivated populations (Fuz, Gan, Nid, and Xia), suggesting artificial bottlenecks. Anthropic cultivation could have linked a naturally occurring discontinuous *F. altissima* population (e.g., due to forest fragmentation) to a continuous population in Indo-Burma and further reduced population differentiation. Therefore, the present distribution patterns and genetic structure of *F. altissima* may have been influenced by human activities. Although, it is difficult to identify the non-confounding factors that exclusively point to the impacts of human influence.

## 5. Conclusions

In conclusion, within Indo-Burma and south China, no clear phylogeographic structure was found in *F. altissima*, presumably as a result of extensive long-distance pollen flow unencumbered by any geographic barrier. Human cultivation and human-mediated dispersal further confound population structure, but it is also likely to have had beneficial consequences for protecting genetic diversity. This large monoecious *Ficus* species with continuous habitat constitutes a single population over a very large area. This lack of phylogeographic structure has been observed in some other plant members of Indo-Burma flora [75]. Such species will probably be highly resilient to global change as they exist over huge surfaces and have likely resisted past change due to large population ranges and a lack of restrictive adaptive specialization to localized conditions.

**Author Contributions:** Conceptualization, J.-F.H., C.T.D. and Y.-Q.P.; methodology, J.-F.H.; software, validation, and formal analysis, J.-F.H.; writing—original draft preparation: J.-F.H.; writing—review and editing, C.T.D. and Y.-Q.P.; visualization, J.-F.H.; supervision, Y.-Q.P.; project administration and funding acquisition, J.-F.H. and Y.-Q.P. All authors have read and agreed to the published version of the manuscript.

**Funding:** This research was supported by the National Natural Science Foundation of China (31800313, 32070487), Yunnan Province Applied Basic Research Project (2019FB034) and the "Light of West China" Program of the Chinese Academic of Sciences to J.-F. Huang.

**Institutional Review Board Statement:** Not applicable.

**Informed Consent Statement:** Not applicable.

**Data Availability Statement:** The data presented in the study are depositing in the NCBI repository, and the accession numbers are shown in the article.

**Acknowledgments:** The authors wish to thank Da-Rong Yang for help in collecting samples; Zhu-Zeng Huang for DNA extraction and PCR amplification; Hong-Hu Meng, Jie Gao and Finn Kjellberg for their advice during the development of this paper.

**Conflicts of Interest:** The authors declare no conflict of interest.

## Appendix A

**Table A1.** Sampling information and genetic parameters of the 37 sampled *Ficus altissima* populations based on cpDNA psbA-trnH and 16 nSSRs.

| Pop. | Country | Site | Lat. (N) | Long. (E) | cpDNA | | | nSSRs | | | | | | |
|------|---------|------|----------|-----------|-------|-------|---------------|-------|-------|-------|--------|-------|-------|-------|
| | | | | | Size | $H_D$ | $\pi$ (10$^{-2}$) | Size | $N_a$ | $N_p$ | $PA_r$ | $H_O$ | $H_E$ | $F_{IS}$ |
| **Yib** | China | Yibin | 28.623 | 104.418 | 1 | na | na | 3 | 2.563 | 1 | 0.063 | 0.396 | 0.427 | 0.269 |
| **Nid** | | Ningde | 26.661 | 119.532 | 6 | 0.000 | 0.000 | 10 | 2.938 | 0 | 0.000 | 0.556 | 0.428 | −0.252 |
| **Pan** | | Panzhihua | 26.554 | 101.680 | 1 | na | na | 1 | 1.438 | 0 | 0.000 | 0.438 | 0.219 | Na |
| **Fuz** | | Fuzhou | 26.153 | 119.291 | 7 | 0.000 | 0.000 | 12 | 2.750 | 0 | 0.000 | 0.469 | 0.435 | −0.033 |
| Liu | | Nujiang | 25.854 | 98.852 | 5 | 0.000 | 0.000 | 5 | 2.625 | 1 | 0.063 | 0.384 | 0.396 | 0.143 |
| **Gan** | | Ganzhou | 25.850 | 114.928 | 9 | 0.000 | 0.000 | 10 | 2.563 | 0 | 0.000 | 0.450 | 0.422 | −0.014 |
| Ten | | Tengchong | 24.943 | 98.387 | 1 | na | na | 1 | 1.375 | 0 | 0.000 | 0.375 | 0.188 | na |
| **Xia** | | Xiamen | 24.447 | 118.062 | 8 | 0.429 | 0.318 | 10 | 4.250 | 3 | 0.188 | 0.494 | 0.548 | 0.151 |
| Mas | | Mangshi | 24.414 | 98.566 | 14 | 0.000 | 0.000 | 14 | 4.125 | 0 | 0.000 | 0.470 | 0.478 | 0.056 |
| Rui | | Ruili | 23.976 | 97.810 | 18 | 0.000 | 0.000 | 18 | 4.000 | 1 | 0.063 | 0.465 | 0.518 | 0.130 |
| Ban | | Lincang | 23.297 | 99.099 | 8 | 0.000 | 0.000 | 8 | 3.500 | 0 | 0.000 | 0.485 | 0.516 | 0.127 |
| Don | | Donggang | 22.889 | 112.281 | 8 | 0.000 | 0.000 | 14 | 3.625 | 1 | 0.063 | 0.518 | 0.494 | −0.011 |
| Loz | | Chongzuo | 22.471 | 107.075 | 6 | 0.000 | 0.000 | 6 | 3.125 | 0 | 0.000 | 0.531 | 0.458 | −0.069 |
| Nah | | Jinghong | 22.131 | 100.675 | 7 | 0.000 | 0.000 | 7 | 3.188 | 2 | 0.125 | 0.509 | 0.474 | 0.008 |
| Yan | | Yangchun | 22.079 | 111.747 | 3 | 0.000 | 0.000 | 8 | 1.313 | 0 | 0.000 | 0.313 | 0.156 | −1.000 |
| Pin | | Pingxiang | 22.056 | 106.736 | 19 | 0.000 | 0.000 | 23 | 5.125 | 3 | 0.188 | 0.519 | 0.528 | 0.040 |
| Che | | Mengla | 21.926 | 101.240 | 11 | 0.000 | 0.000 | 11 | 3.688 | 0 | 0.000 | 0.494 | 0.509 | 0.076 |
| Tai | | Changjiang | 19.317 | 109.027 | 11 | 0.000 | 0.000 | 11 | 3.563 | 2 | 0.063 | 0.542 | 0.512 | −0.011 |
| Jia | | Ledong | 18.716 | 108.832 | 4 | 0.000 | 0.000 | 4 | 2.938 | 0 | 0.000 | 0.469 | 0.451 | 0.104 |
| Dia | | Baoting | 18.666 | 109.914 | 6 | 0.000 | 0.000 | 6 | 3.000 | 1 | 0.063 | 0.465 | 0.405 | −0.057 |
| Man | Burma | Mandalay | 22.549 | 95.998 | 2 | 0.000 | 0.000 | 2 | 1.500 | 1 | 0.063 | 0.469 | 0.242 | −0.875 |
| Nab | | Myotha | 21.697 | 95.635 | 3 | 0.000 | 0.000 | 3 | 2.125 | 0 | 0.000 | 0.479 | 0.396 | −0.011 |
| Law | | Bagan | 21.128 | 94.851 | 17 | 0.000 | 0.000 | 17 | 1.563 | 0 | 0.000 | 0.500 | 0.258 | −0.936 |
| Nya | | Taunggyi | 20.630 | 96.879 | 23 | 0.000 | 0.000 | 23 | 4.313 | 0 | 0.000 | 0.500 | 0.515 | 0.052 |
| Tha | | Thaton | 16.923 | 97.378 | 1 | na | na | 2 | 2.000 | 0 | 0.000 | 0.438 | 0.367 | 0.152 |
| Mou | | Moulmein | 16.433 | 97.658 | 7 | 0.000 | 0.000 | 7 | 2.563 | 0 | 0.000 | 0.469 | 0.369 | −0.197 |
| Cha | Thailand | Lampang | 18.841 | 99.467 | 1 | na | na | 1 | 1.313 | 0 | 0.000 | 0.313 | 0.156 | na |
| Chi | | Chiengmai | 18.795 | 98.962 | 5 | 0.000 | 0.000 | 9 | 3.813 | 2 | 0.125 | 0.500 | 0.495 | 0.048 |
| Tak | | Tak | 16.791 | 98.919 | 1 | na | na | 1 | 1.000 | 1 | 0.063 | 0.063 | 0.031 | na |
| Khy | | Nakhon Nayok | 14.413 | 101.375 | 4 | 0.000 | 0.000 | 4 | 2.000 | 0 | 0.000 | 0.563 | 0.352 | −0.500 |
| Kha | | Rayong | 12.766 | 101.728 | 3 | 0.667 | 0.495 | 3 | 3.250 | 2 | 0.125 | 0.563 | 0.580 | 0.229 |
| Kae | | Kaeng Krachan | 12.538 | 99.478 | 1 | na | na | 1 | 1.500 | 0 | 0.000 | 0.500 | 0.250 | na |
| Hua | | Thap Sakae | 11.625 | 99.615 | 4 | 0.500 | 0.371 | 4 | 3.313 | 3 | 0.188 | 0.365 | 0.540 | 0.451 |
| Pku | | Pkuket | 7.970 | 98.279 | 3 | 0.000 | 0.000 | 3 | 1.500 | 0 | 0.000 | 0.438 | 0.247 | −0.680 |
| Vun | Vietnam | Phu Yen | 12.852 | 109.391 | 1 | na | na | 1 | 1.375 | 0 | 0.000 | 0.375 | 0.188 | na |
| Dal | | Lam Dong | 11.940 | 108.458 | 2 | 0.000 | 0.000 | 2 | 2.375 | 0 | 0.000 | 0.531 | 0.438 | 0.128 |
| Bqk | Cambodia | Bokor | 10.627 | 104.025 | 2 | 0.000 | 0.000 | 2 | 1.500 | 1 | 0.063 | 0.438 | 0.234 | −0.750 |
| Species | | | | | 233 | 0.245 | 0.104 | 267 | 8.500 | na | na | 0.483 | 0.576 | 0.163 |

Note: Bold indicates the six introduced populations collected from non-native range; $H_D$, haplotype diversity; $\pi$, nucleotide diversity; $N_a$, mean number of alleles per locus; $N_p$, number of private alleles; $PA_r$, mean number of private alleles per locus; $H_O$, observed heterozygosity; $H_E$, expected heterozygosity; $F_{IS}$, inbreeding coefficient; na, no available data.

**Table A2.** Null allele frequencies for each locus at each population with more than three individual samples.

| Pop. | Nuclear Microsatellites | | | | | | | | | | | | | | | |
|------|-------|-------|--------|--------|--------|--------|--------|--------|--------|--------|--------|--------|--------|--------|--------|--------|
| | Fal_2 | Fal_9 | Fal_11 | Fal_14 | Fal_28 | Fal_31 | Fal_34 | Fal_41 | Fal_45 | Fal_46 | Fal_48 | Fal_50 | Fal_61 | Fal_62 | Fal_73 | Fal_75 |
| Nid | no | No | No | no | no | no | no | no | no | no | no | no | no | no | no | no |
| Fuz | no | no | **0.279** | no | no | no | no | **0.366** | no | no | no | no | no | no | no | no |
| Liu | no | no | No | no | no | no | no | no | no | no | no | no | no | no | no | no |
| Gan | no | no | **0.304** | no | no | no | no | **0.340** | no | no | no | no | no | no | no | no |
| Xia | no | no | No | no | no | no | no | no | no | no | no | no | no | **0.342** | no | no |
| Mas | no | **no** | **0.298** | no | no | no | no | no | no | no | no | no | no | no | no | no |

**Table A2.** *Cont.*

| Pop. | Nuclear Microsatellites | | | | | | | | | | | | | | | |
| | Fal_2 | Fal_9 | Fal_11 | Fal_14 | Fal_28 | Fal_31 | Fal_34 | Fal_41 | Fal_45 | Fal_46 | Fal_48 | Fal_50 | Fal_61 | Fal_62 | Fal_73 | Fal_75 |
|---|---|---|---|---|---|---|---|---|---|---|---|---|---|---|---|---|
| Rui | no | **0.191** | **0.402** | no | no | no | no | no | no | no | no | no | **0.208** | no | no | no |
| Ban | no | no | **No** | no | no | no | no | no | no | no | no | no | no | no | no | no |
| Don | no | no | **0.423** | no | no | no | no | 0.349 | no | no | no | no | no | no | no | no |
| Loz | no | no | No | no | no | no | no | no | no | no | no | no | no | no | no | no |
| Nah | no | no | No | no | no | no | no | no | no | no | no | no | no | no | no | no |
| Yan | no | no | No | no | no | no | no | no | no | no | no | no | no | no | no | no |
| Pin | no | no | **0.117** | no | no | no | no | **0.228** | no | no | no | no | **0.143** | no | no | no |
| Che | no | no | **0.379** | no | no | no | no | no | no | no | no | no | no | no | no | no |
| Tai | no | no | **0.186** | no | no | no | no | **0.375** | no | no | no | no | no | no | no | no |
| Jia | no | no | No | no | no | no | no | no | no | no | no | no | no | no | no | no |
| Dia | no | no | No | no | no | no | no | no | no | no | no | no | no | no | no | no |
| Law | no | no | No | no | no | no | no | no | no | no | no | no | no | no | no | no |
| Nya | no | no | **0.272** | no | no | no | no | no | no | no | no | no | **0.250** | no | no | **no** |
| Mou | no | no | No | no | no | no | no | no | no | no | no | no | no | no | no | no |
| Chi | no | **0.243** | No | no | no | no | no | **0.296** | no | no | no | no | no | no | no | no |
| Khy | no | no | No | no | no | no | no | no | no | no | no | no | no | no | no | no |

Note: no, indicates no null allele; bold font indicates the deviation from Hardy–Weinberg equilibrium ($p < 0.05$).

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
