# Peer review of "Homogenized Phylogeographic Structure across the Indo-Burma Ranges of a Large Monoecious Fig, Ficus altissima Blume"

_diversity, doi:10.3390/d13120654_

Round 1

Reviewer 1 Report

Dear Editor and Authors,

First of all, I would like to emphasize that I read with great interest this paper that explores the diversity and structure of Ficus altissima populations in Indo-Burma and adjacent south China region using both chloroplast and newly developed nuclear SSR markers. The paper is interesting and easy to read. However, before its acceptance for publication, some things should be clarified.

When the studied species is mentioned for the first time in the title, abstract and text of the main part of the manuscript, the authority, i.e. name of the author of a taxonomic group (genus and scientific names of species) cited after the scientific name should also be specified. The same goes for scientific names of other tree species (Dalbergia cochinchinensis, D. oliveri, Ficus benghalensis, F. binnendijkii, F. curtipes, F. drupacea, F. macrocarpa, F. retusa, F. insipida subsp. indipida, F. sycomrus, and F. sur), genera (Ficus) or even names of bird genera (Megalaima, Dicaeum, and Pycnonotus) mentioned in the paper. The chapters Introduction and Materials and Methods are written clearly. Still, there are several questions and suggestions pertaining to the performed statistical analyses in the chapter Materials and Methods. How did you calculate geographic distances for the Mantel test? Did you try to calculate environmental distances between the studied populations? And to test IBE pattern for nuclear SSR data? PIC values should be calculated for microsatellite markers, as well.

What exactly did you mean by: Dependency on microsites for seed development could result in limited gene flow? Please explain this better.

I would not say that gene flow between populations is strong, given that 18% of total variability is inter-population variability. For woody species this is quite considerable. On the other hand, AMOVA revealed high interpopulation differentiation for cpDNA, suggesting that seed dispersal of F. altissima is limited. On one side you have strong gene flow between populations, and on the other side limited. Please explain better the differences between those two analyses. Please present AMOVA results in a table for a better overview.

A part of the discussion that pertains to cultivated populations should be discussed and explained better. Since when have those populations been cultivated? Why are they cultivated? Why and when did they undergo a through bottleneck event? The chapter on Materials and Methods does not specify why precisely those two models, SMM and TPM, were used?

The part of the discussion in the paragraph between the lines 347 and 354 is not written clearly. The first two sentences have nothing to do with the subsequent three in the same paragraph. That should be explained better. Are there glacial refugia populations in your study?

You report that the studied species grows up to the altitude of 2000 m and that there are no barriers for pollinators? What is the range of altitudes in which the studied species grow? And how come that on such high altitudes there are no barriers to gene flow? What are the distances covered by pollinators? And what are the distances between populations? Does this Ficus species have a continuous habitat where gene flow can take place undisturbed between populations via pollen?

You emphasize that large-scale anthropic cultivation confounds analyses of genetic structure for F. altissima. Still, little is written about the way in which the cultivation of this species affects its diversity. Generally, the part of the paper pertaining to cultivated populations and their relationship with natural populations is not sufficiently explained. That part of the research should be discussed better and more clearly.

The scientific names of species in the list of references should be written in italics.

Kind regards

Reviewer 2 Report

This paper reads very well and the research is to a high standard. I make some minor suggestions in the attached file.

Reviewer 3 Report

The manuscript submitted by Huang et al., is mainly focused on the effect of the human activity on the Indo-Burma biodiversity. For this, authors use a set of chloroplastic (1 cpSSR) and nucleic (16 nSSRs) microsatellite markers. Although there are a high number of species, Ficus could be considered as a key part of the area.

Two panels are included in the figure 1. Both maps with all localizations are proper figures that should be included in Materials and Methods. However, the bar plot from Structure is a proper figure of results. So, this figure should be moved to results.

I do not correctly understand the lines 121-126, I mean, I would like to know how authors check the amplification of the cpSSRs. Do they carry out PCR? Conditions?

Authors should use nSSRs rather than SSRs for the nuclear microsatellite markers.

Why the population Nah was used for the construction of DNA library? More information about the workflow used in the assembling of the transcriptome should be included. The list of 80 primer pairs should be included in Supplementary material.

From my point of view, the use of only one cpSSRs does not give enough information to investigate the origin of this species (no recombination with a haploid genome with uniparental inheritance).

Figure 4a should be bigger considering that names are too small.

Round 2

Reviewer 1 Report

Dear authors,

Thank you for addressing all the concerns/questions raised. I have no further comments to add.

Kind regards

Reviewer 3 Report

N/A